# Cold adaptation in the environmental bacterium *Shewanella oneidensis* is controlled by a J-domain co-chaperone protein network

Nathanael Jean Maillot[1], Flora Ambre Honoré[1], Deborah Byrne[2], Vincent Méjean[1] & Olivier Genest [1]

DnaK (Hsp70) is a major ATP-dependent chaperone that functions with two co-chaperones, a J-domain protein (JDP) and a nucleotide exchange factor to maintain proteostasis in most organisms. Here, we show that the environmental bacterium *Shewanella oneidensis* possesses a previously uncharacterized short JDP, AtcJ, dedicated to cold adaptation and composed of a functional J-domain and a C-terminal extension of 21 amino acids. We showed that *atcJ* is the first gene of an operon encoding also AtcA, AtcB and AtcC, three proteins of unknown functions. Interestingly, we found that the absence of AtcJ, AtcB or AtcC leads to a dramatically reduced growth at low temperature. In addition, we demonstrated that AtcJ interacts via its C-terminal extension with AtcC, and that AtcC binds to AtcB. Therefore, we identified a previously uncharacterized protein network that involves the DnaK system with a dedicated JDP to allow bacteria to survive to cold environment.

[1] Aix Marseille Univ, CNRS, BIP UMR 7281, IMM, 31 Chemin Joseph Aiguier, 13402 Marseille, France. [2] Protein Expression Facility, CNRS, IMM, 31 Chemin Joseph Aiguier, 13402 Marseille, France. Correspondence and requests for materials should be addressed to O.G. (email: ogenest@imm.cnrs.fr)

Every organism possesses a complex network of molecular chaperones to control protein homeostasis, also called proteostasis[1,2]. In stress conditions, the roles of the chaperones become preponderant to prevent the formation of toxic misfolded and aggregated proteins. DnaK in bacteria or Hsp70 in eukaryotes is a major ATP-dependent molecular chaperone playing a key role in the proteostasis network[3–5]. It participates in the folding/unfolding, trafficking, disaggregation, and degradation of a multitude of client proteins, and therefore DnaK is involved in almost all cellular functions, from DNA replication to cell division and metabolism.

DnaK is a two-domain protein containing a nucleotide-binding domain that is connected by a flexible linker to a substrate-binding domain[1,5,6]. The substrate-binding domain is composed of a β-sheet subdomain and an α-helical lid. Allosteric communication between the nucleotide-binding domain and the substrate-binding domain is essential for the chaperone cycle of DnaK[7–9]. When ATP is bound to the nucleotide-binding domain of DnaK, the substrate-binding domain is in an open conformation with the α-helical lid contacting the nucleotide-binding domain[1,8,10]. In this conformation, substrate proteins interact with the substrate-binding domain with low affinity and high exchange rate. In contrast, in the ADP-bound conformation of DnaK, the α-helical lid closes over the substrate in the β-sheet subdomain resulting in high-affinity binding of the substrate and low exchange rate[8,11].

DnaK conformational changes are regulated by two co-chaperones: a J-domain protein (JDP) and a nucleotide exchange factor. The JDP targets DnaK to substrate polypeptides, stimulates DnaK ATP hydrolysis leading to closure of the substrate-binding domain and trapping of the substrate. The nucleotide exchange factor allows ADP to ATP exchange from the ADP-bound DnaK-substrate complex, triggering substrate release[1,8,12].

The JDPs were named from the first extensively studied member of this family, DnaJ from E. coli, and they all possess a J-domain of ~70 amino acids that is essential for the co-chaperone interaction with DnaK[4,5,12,13]. The J-domain is composed of four helices forming a hairpin, and a flexible loop that connects helices II and III which bears the HPD motif, a signature of the JDP proteins. Interestingly, there is a high diversity of JDPs that allows DnaK to possess such a wide variety of functions in the cells, and several JDPs are usually found in a given organism[4,12,13]. The JDPs have been classified into three categories, depending on their domain organization besides the J-domain[12,13]. Class A JDPs possess the structural features of E. coli DnaJ: a J-domain followed by a G/F rich region, two similar β-barrel domains containing a Zinc finger, and a C-terminal domain allowing dimerization. Class B is similar to class A, except that the Zinc finger region is missing. Class C represents the most diverse category of JDPs since these proteins possess a conserved J-domain that is associated with various domains different from that of class A and B. In some examples, these additional domains are responsible of targeting specific substrates to DnaK, they can allow the JDPs to localize at a specific position in the cell therefore helping the recruitment of DnaK at this position, and many are still of unknown functions[12,13]. Therefore, although many JDPs have been studied, much more work is required to understand the functional diversity and molecular mechanisms of many class C JDPs, in particular for bacteria which contain a great reservoir of uncharacterized JDPs[14].

Bacteria from the Shewanella genus are ubiquitous aquatic γ-proteobacteria found in marine and fresh waters, and in sediments[15]. Given their natural ecological niches, these bacteria possess the ability to adapt to many stress conditions, including variations in a vast range of temperatures (from 4 °C to more than 42 °C), in hydrostatic pressure, or in osmotic conditions.

Remarkably, Shewanella oneidensis appears like an exquisite bacterial model for chaperone study. Indeed, we have recently shown that Hsp90, another major chaperone, plays a crucial role for the growth of S. oneidensis at high temperature[16,17], although Hsp90 is expendable in Escherichia coli[18]. By exploring the genome of S. oneidensis, we found that it encodes a functional DnaK system composed of the DnaK chaperone, the nucleotide exchange factor GrpE, and several JDPs, including DnaJ. Indeed, we identified four putative uncharacterized class C JDPs. Interestingly, S. oneidensis seems to have a strong requirement for the DnaK system since dnaK and dnaJ genes cannot be deleted[19].

In this paper, we show that the DnaK system driven by a JDP, AtcJ, is required for cold adaptation in S. oneidensis. In addition, we found that AtcJ interacts with another protein of unknown function, AtcC, encoded in the same operon and that this protein is also essential during cold stress. Finally, we found that AtcC forms a complex with AtcB showing that a network of interaction allows the bacteria to cope with cold environments.

## Results

**AtcJ is a functional J-domain protein.** In addition to the canonical class A JDP DnaJ and the two class C JDPs DjlA and HscB[4], we found that the genome of S. oneidensis encodes four putative uncharacterized class C JDPs. Here, we focus on one of them, SO_1850, that we called AtcJ (Adaptation to cold protein J). AtcJ is a 94-amino acid protein with a molecular mass of 11 kDa. As for all proteins of the Jdp family, it possesses a conserved J-domain that is composed of four helices and contains the classical tripeptide $_{31}HPD_{33}$ located on an exposed loop connecting helices II and III (Fig. 1a, b)[4,5,12]. The J-domain of AtcJ shares 63% of sequence similarity with that of DnaJ from E. coli. Besides the J-domain, AtcJ contains a 21 amino acid C-terminal extension of unknown function only found in the proteins homologous to AtcJ (Supplementary Fig. 1A).

To test if the J-domain of AtcJ was functional, we constructed a chimera between E. coli DnaJ (DnaJ$_{Ec}$) and AtcJ by substituting the J-domain of DnaJ$_{Ec}$ by that of AtcJ (Fig. 1c), and we took advantage of the known phenotypes (high-temperature sensitivity and motility defects) of the Δ3 E. coli strain devoid of three DnaK co-chaperones (DnaJ, CbpA, and DjlA)[20–22]. As expected, the growth of the Δ3 strain containing an empty vector was strongly reduced at high temperature, 43 °C, compared with 37 °C and production of DnaJ$_{Ec}$ from a plasmid rescued this phenotype (Fig. 1d)[21]. Interestingly, production of the AtcJ chimera allowed growth of the Δ3 strain at 43 °C, indicating that the J-domain of AtcJ is functional and can replace that of DnaJ$_{Ec}$ (Fig. 1d). Similarly, production of the AtcJ chimera restored the absence of motility of the Δ3 strain, although at a lower level than DnaJ$_{Ec}$ (Fig. 1e).

It is known that a single H to Q mutation in the conserved HPD motif of the J-domain proteins prevents physical and functional interactions with DnaK[23,24]. The chimera was therefore mutated accordingly (H31Q) (Fig. 1c), and as expected, production of this mutated chimera did not restore the growth defect at high temperature and the loss of motility of the Δ3 strain (Fig. 1d, e). As a control, we checked that the mutated chimera was produced at the same level as the wild-type chimera (Supplementary Fig. 1B). These experiments indicate that the J-domain of AtcJ can functionally replace that of DnaJ, therefore allowing association with DnaK.

Binding of JDPs to DnaK is known to trigger allosteric changes in DnaK that result in ATPase activity stimulation[7,12], and as a control, we found that DnaJ from S. oneidensis stimulated the ATPase activity of DnaK (Fig. 1f; Supplementary Fig. 1C). To clearly demonstrate that AtcJ is a functional JDP, we measured

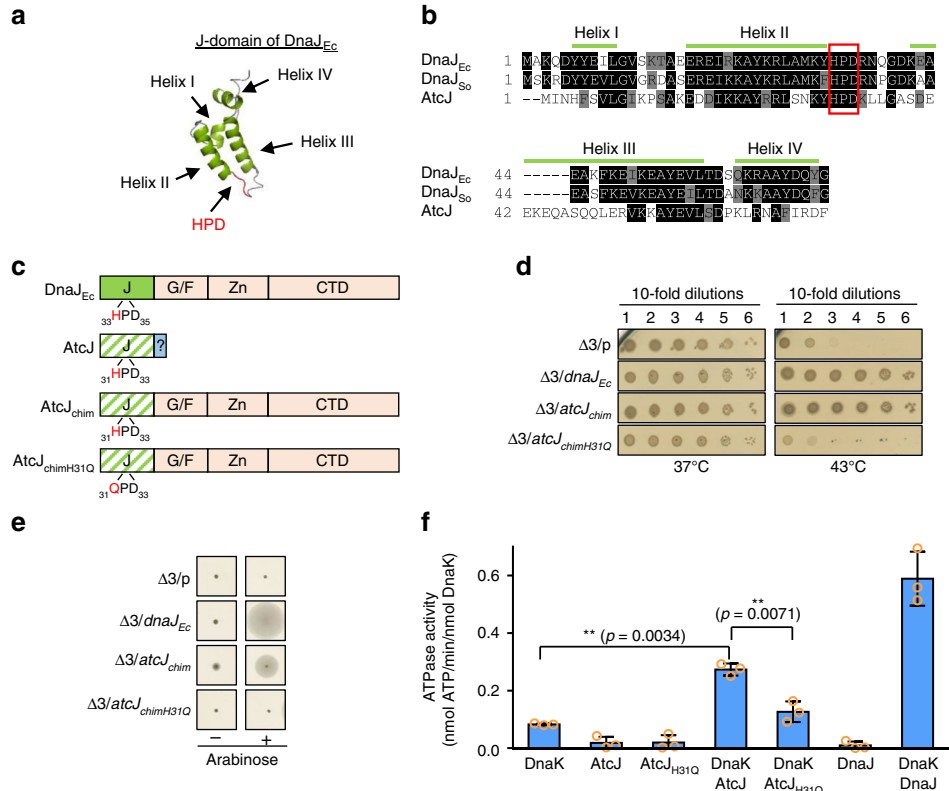

**Fig. 1** AtcJ is a functional J-domain protein. **a** Structure of the J-domain of *E. coli* DnaJ (PDB: 1XBL). Helices are shown in green, loops are in gray, and the conserved HPD tripeptide is in red. **b** Sequence alignment of the J-domains of *E. coli* DnaJ (DnaJ$_{Ec}$), *S. oneidensis* DnaJ (DnaJ$_{So}$), and AtcJ using the ClustalW/Omega method and Boxshade (ExPaSy). Black boxes indicate that the residues are strictly conserved and gray boxes that they have conserved substitutions. Green lines indicate the helices of DnaJ$_{Ec}$ shown in **a**. The HPD motif is framed in red. **c** Schematic of the chimera. The J-domain of DnaJ$_{Ec}$ is shown in green, and the glycine–phenylalanine-rich domain (G/F), the zinc-binding domain (Zn), and the C-terminal domain (CTD) are shown in light orange. The J-domain (J) of AtcJ is green hatched, and the C-terminal extremity of 21 residues is colored blue. The AtcJ chimera (AtcJ$_{chim}$) possesses the J-domain of AtcJ and the G/F, Zn and CTD domains of DnaJ$_{Ec}$. The AtcJ$_{chimH31Q}$ chimera has the H31Q point mutation in the J-domain of AtcJ. **d** Production of the wild-type chimera (AtcJ$_{chim}$) suppresses the growth phenotype of the *E. coli* Δ3 strain (*dnaJ⁻ cbpA⁻ djlA⁻*) at high temperature. *E. coli* Δ3 strain containing the pBad33 vector (p) or plasmids as indicated were grown at 28 °C. Ten time serial dilutions were spotted on LB-agar plates containing 2% L-arabinose (w/v). Plates were incubated overnight at 37 °C and 43 °C. **e** Production of the wild-type chimera (AtcJ$_{chim}$) partially suppresses the motility phenotype of the *E. coli* Δ3 strain (*dnaJ⁻ cbpA⁻ djlA⁻*). Same strains as in **d** were spotted on LB soft agar (0.3% (w/v)), and motility was observed by the circumference halo around the initial spot after 1 day at 28 °C. In **d** and **e**, plates are representative of at least three experiments. **f** Stimulation of the DnaK ATPase activity by AtcJ. ATPase activities were measured at 37 °C with 10 μM DnaK, 50 μM AtcJ or AtcJ$_{H31Q}$, or 2 μM DnaJ where indicated. The data from three replicates are shown as mean ± SD. *t* Test analysis shows that the difference measured is significant (**$p < 0.01$)

DnaK ATPase activity in the presence of full-length AtcJ. We observed that, although at a lower level than with DnaJ, the ATPase activity of DnaK was stimulated 3.3 times by wild-type AtcJ, whereas very low stimulation was measured in the presence of AtcJ$_{H31Q}$ (Fig. 1f). In the presence of the nucleotide exchange factor GrpE from *S. oneidensis*, the ATPase activity of DnaK was increased, and a 2.7-fold stimulation was observed when AtcJ was added (Supplementary Fig. 1C). Here again, AtcJ$_{H31Q}$ did not almost stimulate the ATPase activity of DnaK in the presence of GrpE (Supplementary Fig. 1C).

Altogether, these experiments indicate that AtcJ possesses a bona fide J-domain that allows physical and functional binding with the DnaK chaperone.

**AtcJ is essential for cold resistance**. To understand the role of AtcJ in *S. oneidensis*, we compared the growth of an *atcJ* deletion strain with a wild-type strain. The two strains were grown at 28 °C, a permissive temperature for *S. oneidensis*, and dilutions were spotted on plates that were subsequently incubated at several temperatures. Strikingly, we found that the growth of the *atcJ*

deletion strain was dramatically reduced at low temperature, 7 °C, compared with the wild-type strain that grew well after a few days (Fig. 2a). Similar phenotypes were also observed at temperatures ranging between 5 °C and 10 °C. At 15 °C and 28 °C, no relevant growth difference was observed between the two strains, as well as at 35 °C, a heat-stress temperature for *S. oneidensis* (Fig. 2a)[16].

We also compared the growth of the wild-type and the *atcJ* deletion strains in liquid cultures. Interestingly, although the wild-type strain grew well at 7 °C, growth of the *atcJ* deletion strain was strongly impaired at this temperature (Fig. 2b). At 28 °C, the two strains grew similarly as expected (Fig. 2b).

To confirm that AtcJ is essential at low temperature for *S. oneidensis* growth, *atcJ* was cloned into an inducible vector, and the resulting plasmid was introduced in the *atcJ* deletion strain. We observed that, at 7 °C, the production of AtcJ from a plasmid rescued the growth defect of the *atcJ* deletion strain to a level similar to that of the wild-type strain containing an empty vector (Fig. 2c). Interestingly, we found that the production of the AtcJ$_{H31Q}$ mutant that is affected for DnaK stimulation (Fig. 1f) did not allow growth of the *atcJ* deletion strain at low temperature

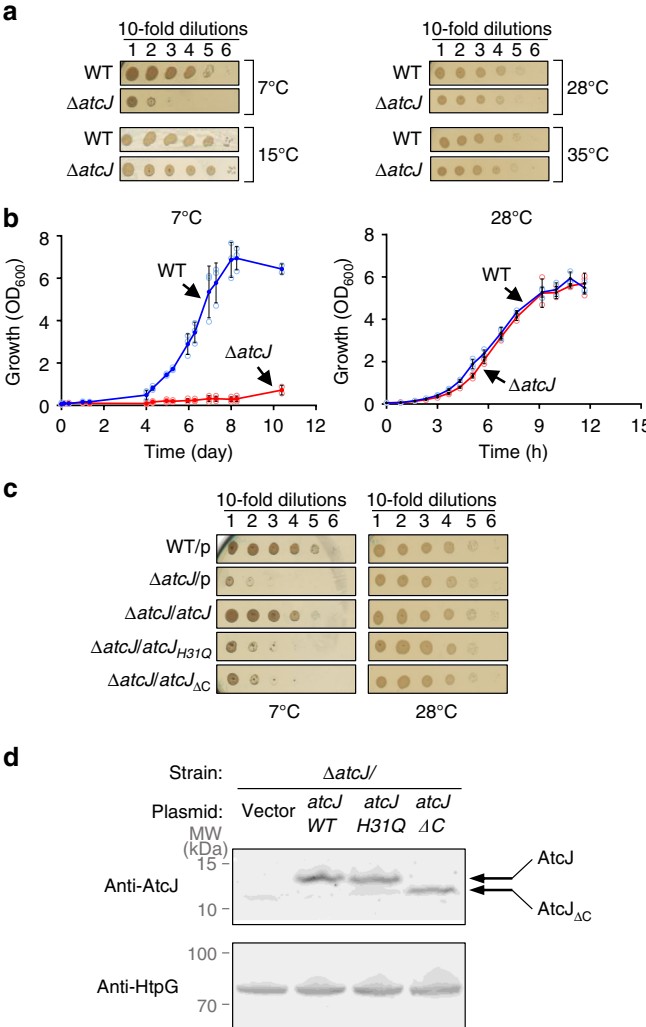

**Fig. 2** AtcJ allows cold adaptation. **a** Growth of *S. oneidensis* wild-type and mutant (*ΔatcJ*) strains at several temperatures on LB-agar plates. After initial growth at 28 °C, strains were diluted to OD$_{600}$ = 1, and 10-time serial dilutions were spotted on LB-agar plates. Plates were incubated 10 days at 7 °C, 5 days at 15 °C, and 1 day at 28 °C and 35 °C. **b** Growth of *S. oneidensis* wild-type and mutant (*ΔatcJ*) strains at 7 °C and 28 °C in liquid media. After initial growth at 28 °C, strains as in A were diluted to OD$_{600}$ = 0.05, and were incubated at 7 °C or 28 °C with shaking. Absorbance was measured with time. The data from three replicates are shown as mean ± SD. **c** Complementation of the growth phenotype. Wild-type or *ΔatcJ* strains containing as indicated the pBad33 vector (p) or the plasmids producing wild-type AtcJ, AtcJ$_{H31Q}$ or AtcJ$_{ΔC}$ that lacks the last 21 residues were treated as in **a**, and dilutions were spotted on agar plates containing 0.2% arabinose. Plates were incubated at 7 °C (10 days) or at 28 °C (1 day). In **a** and **c**, plates are representative of at least three experiments. **d** The *ΔatcJ* strains containing the same plasmids as in **c** were grown overnight at 28 °C with 0.2% arabinose. Protein extracts were analyzed by western blot using anti-AtcJ antibody. The upper arrow indicates the bands corresponding to AtcJ-WT or H31Q mutant, and the lower arrow AtcJ$_{ΔC}$. The HtpG protein (lower panel) was detected by a specific antibody to show that the same amount of the different extracts was loaded on the gel

(Fig. 2c). We checked by western blot that AtcJ$_{H31Q}$ was produced in the same amount as the wild-type AtcJ (Fig. 2d). These results strongly suggest that the essential function of AtcJ to support *S. oneidensis* growth at low temperature requires a functional J-domain as expected for a DnaK co-chaperone.

As described above, AtcJ is a very short Jdp that only contains 21 amino acids in addition to its J-domain. Using the growth phenotype of the *atcJ* deletion strain, we addressed the importance of its C-terminal region by producing a truncated mutant protein, AtcJ$_{ΔC}$ that lacks the last 21 amino acids. We found that AtcJ$_{ΔC}$ did not rescue the slight growth of the *atcJ* deletion strain at low temperature, indicating that the C-terminal extension of AtcJ is also essential for its function (Fig. 2c, d).

**atcJ is encoded in an operon involved in cold resistance.** By analyzing the genome of *S. oneidensis*, we noticed that a predicted three-gene operon of unknown function (SO_1849, SO_1848, and SO_1846) that we called *atcA*, *atcB*, and *atcC*, respectively, was encoded 120-bp downstream of *atcJ*. Comparison of 80 bacterial genomes from the *Shewanella*, *Moritella*, and *Aeromonas* genus indicates that the synteny between *atcJ* and the three other genes is conserved, and that, in the majority of bacterial genomes (74/80), only a few nucleotides (8–24 bp) separated the end of *atcJ* from the beginning of the following gene of the putative operon. This observation suggests that, in *S. oneidensis*, *atcJ* could be encoded in the same operon as the three other genes. This hypothesis was explored by RT-PCR using converging primers located in each of the four genes (Fig. 3a). After RNA retro-transcription, PCR amplifications were observed between *atcJ* and *atcA*, *atcA* and *atcB*, and *atcB* and *atcC* (Fig. 3b), meaning that a same transcript contains several genes. *atcJ* was amplified from cDNA as a positive control (Fig. 3b, lane 1), whereas no ampli-fication was observed using mRNA as matrix with *atcJ* primers (Fig. 3b, lane 5). These experiments strongly suggest that *atcJ*, *atcA*, *atcB*, and *atcC* are part of the same operon.

The proteins AtcA (20 kDa), AtcB (29 kDa), and AtcC (34 kDa) from *S. oneidensis* as well as their homologs in other organisms have not yet been characterized. Classical bioinfor-matics analysis did not identify any known domains, protein sequence motifs, signal peptide or transmembrane segment. Since we showed that AtcJ is important for cold resistance, we wondered whether the other proteins encoded in the same operon could also play a role in this resistance mechanism. Wild-type and *atcA*, *atcB*, or *atcC* deletion strains were grown overnight at permissive temperature, diluted and spotted on plates that were incubated at 7 °C or 28 °C. We found that although the *atcA* deletion strain grew like the wild-type, the *atcB*, and *atcC* deletion strains had very poor growth at 7 °C and behave similarly to the *atcJ* deletion strain (Figs. 2a, 3c). At 28 °C, all the strains grew as the wild-type. Growth defects at low temperature of the *atcB* and *atcC* deletion strains were complemented by the production from a plasmid of AtcB and AtcC proteins, respectively, in the presence of the arabinose inducer (Fig. 3d). These results support the hypothesis that AtcJ, AtcB, and AtcC together work in the same pathway involved in cold resistance. However, growth defects of the *atcJ*, *atcB*, or *atcC* strains at low temperature were not rescued by the production of the other Atc proteins suggesting a specific role for these proteins (Supplementary Fig. 2).

Given the importance of AtcJ, AtcB, and AtcC at low temperatures, we tested if the expression of the *atcJABC* operon is induced at low temperature. Wild-type *S. oneidensis* cells were grown at 7 °C, 28 °C, and 37 °C, RNA was isolated, reverse transcribed in cDNA, and tested by quantitative PCR. We found that expression of the *atcJABC* operon did not almost vary with temperatures, although a very slight increase of expression was measured at low temperature (Supplementary Fig. 3A). In contrast, the expression of *dnaK* was strongly increased at 37 °C as expected[25,26], and interestingly we observed that it was reduced at 7 °C (Supplementary Fig. 3B).

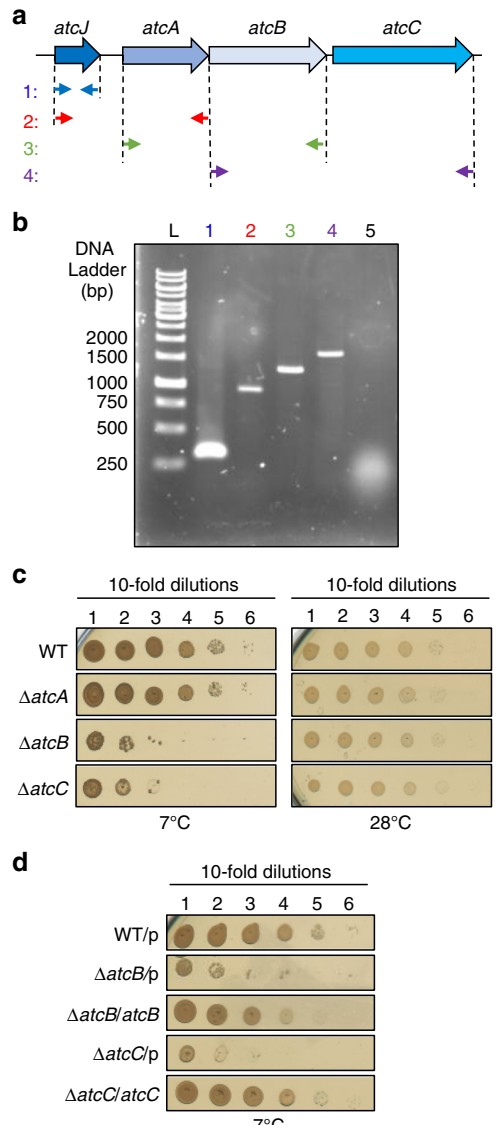

**Fig. 3** *atcJ* is encoded in an operon crucial for cold resistance. **a** Schematic of the *atc* operon. Specific oligonucleotides indicated by little arrows were used to amplify DNA fragments from the cDNA matrix. The fragments correspond to *atcJ* (1), *atcJ-atcA* (2), *atcA-atcB* (3), and *atcB-atcC* (4). **b** cDNA amplification. The total RNA from wild-type *S. oneidensis* strain grown at 7 °C to exponential phase was extracted and retro-transcribed in cDNA. PCR were performed using cDNA as a matrix and the oligonucleotides indicated in **a**. No amplification (lane 5) was observed when the primers 1 (see panel **a**) were used in the PCR with RNA as a matrix, indicating that the RNA preparation was not contaminated by the chromosome. **c** Growth of the Δ*atcA*, Δ*atcB*, and Δ*atcC* mutant strains. After initial growth at 28 °C, *S. oneidensis* strains were diluted to OD$_{600}$ = 1, and 10-time serial dilutions were spotted on LB-agar plates. Plates were incubated 10 days at 7 °C and 1 day at 28 °C. **d** Complementation of the growth phenotype. Wild-type, Δ*atcB*, or Δ*atcC* strains containing as indicated the pBad33 vector (p) or plasmids with the *atcB* or *atcC* gene were treated as in **c**, and dilutions were spotted on 0.2% arabinose LB-agar plates that were incubated at 7 °C for 10 days. In **c** and **d**, plates are representative of at least three experiments

**AtcJ interacts with AtcC**. To better understand the role of AtcJ, we wondered if it could interact with a protein from the *atc* operon. We first used a bacterial two-hybrid assay based on the reconstitution of the T18 and T25 catalytic domains of the

adenylate cyclase of *Bordetella pertussis*[27,28]. When AtcJ fused to the T18 domain and AtcC fused to the T25 domain were produced together in an adenylate cyclase deleted *E. coli* strain, we measured a high level of β-galactosidase activity which reflects the interaction between AtcJ and AtcC (Fig. 4a). This interaction is relevant since production of T18-AtcJ with the T25 domain alone, or T25-AtcC with the T18 domain alone led to background levels of β-galactosidase activity. In contrast, no interaction was detected between AtcJ and AtcA or AtcB, indicating that AtcC could be the only partner of AtcJ in the *atc* operon (Fig. 4a).

Since we found that AtcC as well as the C-terminal extension of AtcJ are both essential to support cold resistance (Figs. 2c, 3c), we wondered if this region could be involved in AtcC binding. To answer this point, we constructed two mutants of AtcJ fused to the T18 domain, one lacking the last 21 amino acids, AtcJ$_{\Delta C}$, and the other producing only the last 21 amino acids, AtcJ-C. Strikingly, we observed that binding with AtcC was abolished with the mutant AtcJ$_{\Delta C}$, whereas an interaction was measured with AtcJ-C (Fig. 4a). These results support the idea that the C-terminal extension of AtcJ is essential for AtcC binding, and that the J-domain is not involved in this interaction.

To confirm the interaction between AtcJ and AtcC and to have more insight into the affinity and thermodynamic parameters, we performed isothermal titration calorimetry using purified AtcC with wild-type AtcJ, AtcJ$_{\Delta C}$, or a synthetic peptide corresponding to the last 21 amino acids of AtcJ that we called pep21. We found that wild-type AtcJ strongly interacted with AtcC with a Kd of 7 nM ± 2 and a stoichiometry ratio of 1 (Fig. 4b), whereas no binding was detected with the AtcJ$_{\Delta C}$ mutant (Fig. 4c). Moreover, the synthetic peptide pep21 interacted with AtcC with a Kd of 80 nM ± 12, confirming that the C-terminal region of 21 amino acids of AtcJ is the main site of interaction with AtcC (Fig. 4d). As a control, we found that a scrambled peptide did not interact with AtcC showing that AtcC interacts specifically with the last 21 amino acids of AtcJ (Supplementary Fig. 4).

We then developed a thermal shift assay to monitor the stability of AtcC in the presence or the absence of the peptide corresponding to the last 21 residues of AtcJ. In this experiment, purified AtcC was analyzed in a thermocycler with increasing temperature in step-by-step mode to induce thermal unfolding of the protein. A specific fluorophore, the Sypro orange, is quenched in aqueous solution and becomes unquenched when it interacts with exposed hydrophobic regions of the protein during unfolding. Therefore, the increase in fluorescence reflects the unfolding of AtcC. We found that AtcC alone had a melting temperature (Tm) of 31 °C ± 0.5 (Fig. 4e). Strikingly, when AtcC was incubated with the peptide, the Tm shifted to 44.5 °C ± 0.5 (Fig. 4e). Similar results were also observed when full-length AtcJ was incubated with AtcC, the Tm shifted to 44 °C ± 0.5 (Fig. 4e). We were not able to accurately determine a Tm value for AtcJ alone because its unfolding gave a very weak fluorescent signal for unknown reasons (Fig. 4e).

Together, these results indicate that binding of the last 21 amino acids of AtcJ stabilizes AtcC, either by directly contacting a region prone to unfolding, or by triggering a conformational change in AtcC that becomes more resistant to thermal unfolding.

**AtcC binds to AtcB**. Our data show that AtcJ, AtcB, and AtcC, that are encoded in the same operon, are functionally related to cold resistance in *S. oneidensis*, and that AtcJ interacts with AtcC. We next wondered if AtcA, AtcB, and AtcC are physically connected. Using a two-hybrid assay, we found that AtcB and AtcC interacted (Fig. 5a). However, no interaction was observed between AtcA and the two other proteins (Fig. 5a).

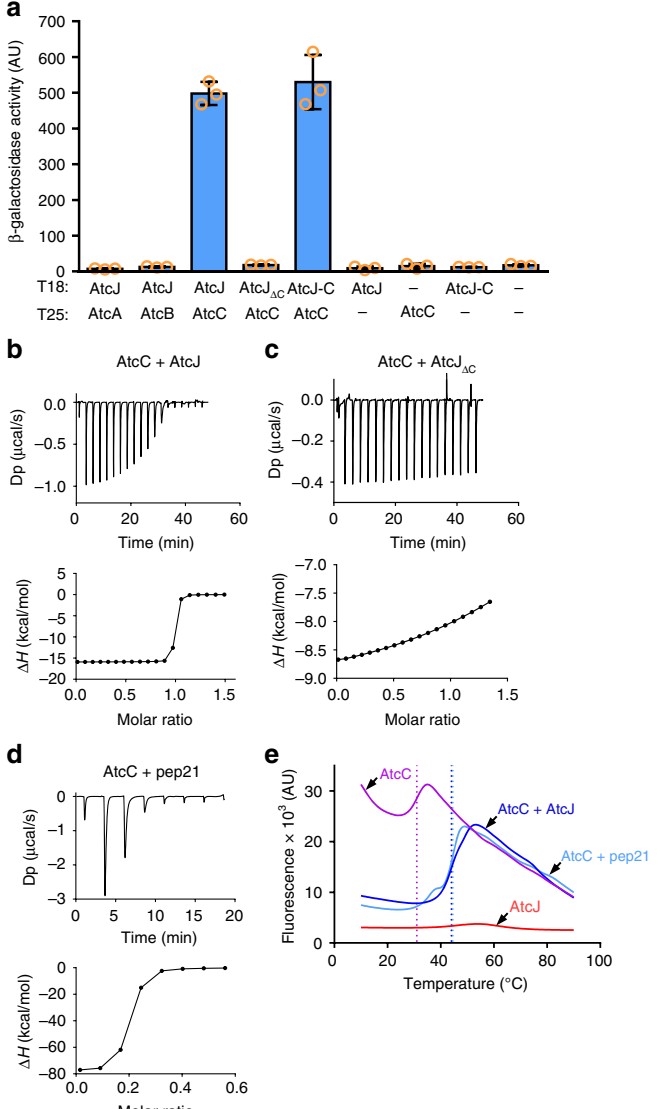

**Fig. 4 AtcJ interacts with AtcC. a** Bacterial two-hybrid assay showing that AtcJ and AtcC interact. *E. coli* Bth101 strain co-transformed as indicated with the T18 and T25 plasmids coding for protein fusion between T18 and AtcJ, AtcJ$_{\Delta C}$, or the last 21 amino acids of AtcJ (AtcJ-C), and protein fusion between T25 and AtcA, AtcB, or AtcC were grown overnight at 28 °C. β-galactosidase activity was measured as explained in the Methods section. The data from three replicates are shown as mean ± SD. **b–d** ITC experiments. Binding assays were performed at 25 °C with 36 μM AtcC and 285 μM AtcJ (**b**), 36 μM AtcC and 285 μM AtcJ$_{\Delta C}$ (**c**), or 36 μM AtcC and 285 μM of the pep21 peptide corresponding to the last 21 amino acids of AtcJ (**d**). Top panels show heat exchange upon ligand titration, and bottom panels show the integrated data with binding isotherms fitted to a single-site binding model. The data shown are representative of two independent experiments. **e** Thermal Shift Assay experiments were performed as indicated in Methods with 10 μM AtcC, 40 μM AtcJ, or 40 μM pep21. Proteins were mixed as indicated in the figure, SYPRO Orange was added, and the temperature was increased from 10 °C to 90 °C with a 0.5 °C step size. Fluorescence was measured at each temperature. Dot lines indicate the temperature melting point of AtcC alone (purple, 31 °C ± 0.5), with AtcJ (blue, 44 °C ± 0.5), or with pep21 (light blue, 44.5 °C ± 0.5). The data shown are representative of four independent experiments

The interaction between AtcB and AtcC was also confirmed by a co-purification experiment. Plasmids allowing the production of AtcC with a 6His-tag and AtcB with a CBP tag, as well as control plasmids, were co-transformed in an *E. coli* strain and the

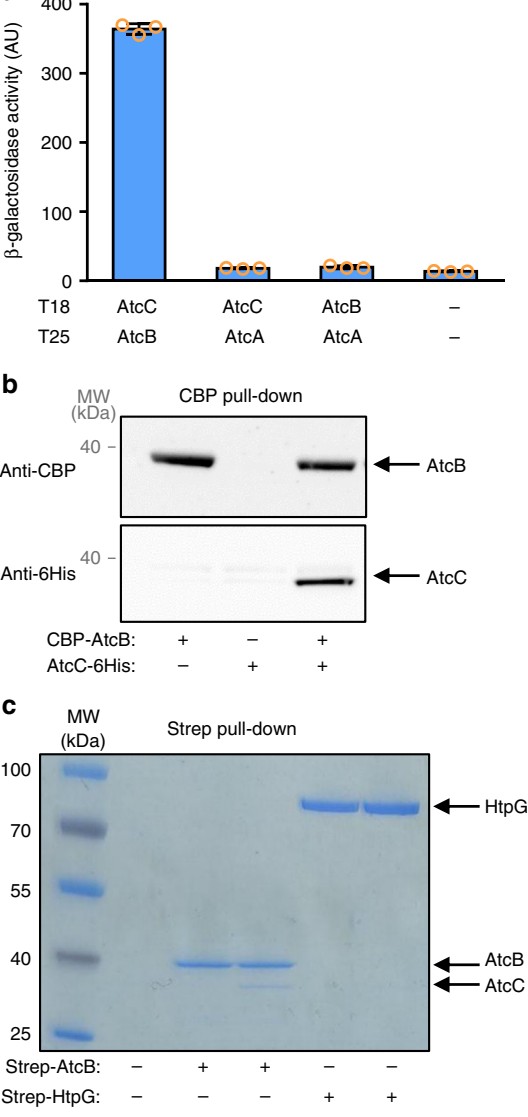

**Fig. 5 AtcB interacts with AtcC. a** Bacterial two-hybrid assay. *E. coli* Bth101 strain co-transformed as indicated with the T18 and T25 plasmids coding for protein fusion between T18 and AtcC or AtcB, and protein fusion between T25 and AtcA or AtcB were grown overnight at 28 °C. β-galactosidase activity was measured as explained in the Methods section. The data from three replicates are shown as mean ± SD. **b** Co-purification assay. AtcB with a CBP tag was produced with or without AtcC with a 6-His tag in the *E. coli* MG1655 strain. CBP-AtcB was purified using CBP affinity resin and bound proteins were analyzed by western blot with anti-CBP and anti-His antibodies as indicated. **c** In vitro co-purification assay with purified proteins. Purified AtcB (16 μg) or HtpG (16 μg) with a Strep-tag, and purified AtcC (100 μg) with a 6His-tag were mixed as indicated (final volume 200 μL). Proteins with a Strep-tag were pulled down on Strep-Tactin beads, washed several times, and proteins were separated on SDS-PAGE. The results shown in **b** and **c** are representative of three independent experiments

proteins were produced by adding the arabinose inducer. We purified AtcB on calmodulin beads and observed on a western blot with a 6His antibody that AtcC was co-purified with AtcB (Fig. 5b).

To make sure that the interaction between AtcB and AtcC was direct, in vitro co-purification experiments using purified proteins

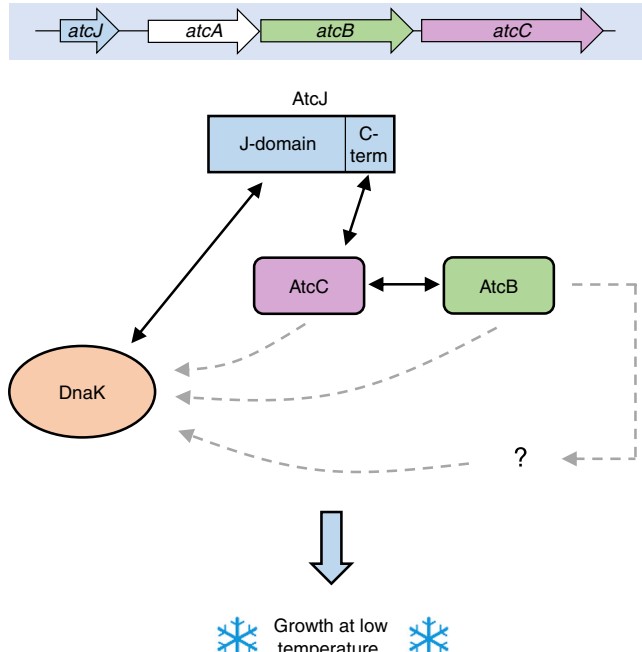

**Fig. 6** Model of the role of the Atc proteins in cold resistance. The AtcJ, AtcA, AtcB, and AtcC proteins are encoded from the *atcJABC* operon. AtcJ binds DnaK with its J-domain, and binds AtcC with its last 21 amino acids (C-term). In addition, AtcC interacts with AtcB. This network of interaction could allow the transfer of AtcC, AtcB, or another protein to DnaK, or could target the DnaK system to a specific location in the cell. Altogether, these proteins play a key role to support bacterial growth at low temperature. Black solid arrows indicate physical or functional interactions demonstrated in this study; gray dashed arrows indicate putative interactions

were performed. AtcB with a Strep-tag and AtcC with a 6His-tag were incubated together and AtcB was pulled down on Strep-Tactin beads. Interestingly, we observed that AtcC co-purified with AtcB (Fig. 5c), confirming that the binding is direct. As negative controls, we showed that AtcC neither interacted with the Strep-Tactin beads nor interacted with another protein with a Strep-tag, HtpG (Fig. 5c).

Altogether, our results indicate that there is a network of interactions between proteins encoded in the *atc* operon. We found that AtcC interacts with AtcJ and with AtcB, and that together these proteins support cold resistance.

## Discussion

In this paper, we identified a previously uncharacterized JDP, AtcJ, which is essential for the growth of *S. oneidensis* at low temperature. *atcJ* is the first gene of a 4-gene operon (*atcJABC*) that is involved in cold adaptation since the absence of *atcB* or *atcC* resulted in the same cold-sensitive phenotype as the Δ*atcJ* strain. Interestingly, a network of interaction was identified between proteins encoded from this operon: AtcJ interacts with AtcC, and AtcC interacts with AtcB (Fig. 6). In addition, discrete regions of AtcJ were identified for DnaK binding (i.e., the J-domain), and for interaction with AtcC (i.e., the C-terminal extremity of 21 amino acids) (Fig. 6).

Our experiments strongly suggest that AtcJ works as a co-chaperone of the DnaK system to support cold resistance. Indeed, AtcJ possesses a bona fide J-domain that can replace that of DnaJ in complementation assays of the characterized phenotypes of an *E. coli* strain devoid of JDPs (Fig. 1d, e). In addition, AtcJ stimulated the ATPase activity of DnaK (Fig. 1f), whereas AtcJ$_{H31Q}$ that has the well-characterized single point mutation in the highly

conserved HPD motif that prevents DnaK binding did not activate the ATPase (Fig. 1f). Finally, the production of AtcJ$_{H31Q}$ did not support growth of the Δ*atcJ* strain at low temperature (Fig. 2c). However, it is important to mention that the stimulation of the ATPase activity of DnaK was much higher with DnaJ than with AtcJ (Fig. 1f; Supplementary Fig. 1C). This result was expected because in addition to the J-domain, other domains of DnaJ (absent in AtcJ) including the G/F region participate in the stimulation[23,24,29].

Based on our data, we propose that AtcJ, by contacting both the DnaK chaperone by its J-domain and AtcC by its C-terminal extremity could connect DnaK to AtcC or to the complex formed by AtcC and AtcB (Fig. 6). AtcC or AtcB might then be transferred to DnaK as substrates, although the quite high binding affinity (7 nM) we determined for AtcJ and AtcC (Fig. 4b) seems not to be in agreement with a transient interaction expected between a co-chaperone and its substrate[30,31]. In other hypotheses, the AtcJ-AtcC-AtcB proteins could help in the recruitment of DnaK substrates, or target DnaK to a specific location in the cell. Altogether, the Atc proteins by a still unknown mechanism play a key role in supporting cold adaptation.

We found that the *atcJABC* operon is constitutively expressed from 7 °C to 37 °C (Supplementary Fig. 3A), although we showed that AtcJ, AtcB, and AtcC are crucial for bacterial growth only at low temperature (Figs. 2a, 3c). One possibility is that these proteins have to be already synthesized to allow cell survival to a sudden cold shock. On the other hand, these proteins might play an alternative role in addition to cold adaptation.

The operonic organization of AtcJ suggests that AtcJ might have evolved to be dedicated to work with AtcC or AtcB or to a protein targeted by the AtcB–AtcC complex, in contrast to DnaJ that interacts with many substrate proteins. This is consistent with other class C JDP that have selective client binding[12,13]. For example, the class C JDP HscB (Jac1 in mitochondria) that is conserved from bacteria to eukaryotes interacts with IscU (Isu), a scaffold protein involved in Fe–S cluster assembly[32]. Interestingly, HscB is encoded in the same operon as IscU in bacteria. HscB transfers IscU to a DnaK homolog, HscA (Ssq1), that facilitates the loading of Fe–S clusters into the recipient apo-proteins. Another example is given by eukaryotic *sc*Swa2 (*hs*DNAJC6) that specifically binds clathrin proteins coated on vesicles, and with Hsp70 dissociate clathrin from vesicles, a process important to control membrane traffic pathways[33].

Some class C JDPs are also known to recruit Hsp70 at specific locations in the cell like for instance Zuo1 in *S. cerevisiae* at the ribosome or Pam18 at mitochondrial translocons[12,13]. Therefore, an alternative model would be that the complex AtcB-AtcC might localize AtcJ, and therefore DnaK at some particular location where it might be needed to cope with cold environments. This hypothesis is in agreement with the reduced expression of *dnaK* observed at low temperature (Supplementary Fig. 3B) that could potentially lead to DnaK scarcity. Therefore, AtcJ could locally concentrate DnaK where it is required.

In bacteria, low temperature is a stress known to affect several cellular components and pathways, including a decrease in membrane fluidity, stabilization of RNA secondary structures leading to reduced transcription and translation, a diminution in protein export, and a slowdown in protein folding[34–37]. Bacterial cells have therefore developed ways to counteract these adverse effects. For example, they modify their membrane composition to maintain viscosity[34], and some *Shewanella* species produce a specific polyunsaturated fatty acid called eicosapentaenoic acid[38]. In addition, bacteria adapt their RNA metabolism by producing a set of cold-shock proteins (Csp) and other proteins such as RNA helicase and exoribonucleases[34,39,40]. Chaperone proteins are also important players to cope with cold conditions and favor the

folding of substrates. For example, the level of the trigger factor (TF), a chaperone that interacts with the ribosome, increases in *E. coli* and in some psychrophilic bacteria when temperature decreases[41,42]. Moreover, the absence of DnaK or its three co-chaperones together[20,43], or of the export chaperone SecB[44] lead to cold-sensitive phenotypes in *E. coli*, indicating that these chaperones are involved in protein homeostasis at low temperature. Recently, it has been shown that cold conditions could induce prophage excision in *S. oneidensis* leading to inhibition of trans-translation, a mechanism important for protein synthesis quality control, and consequently promoting biofilm formation at low temperatures[45].

Future work will aim at understanding if the Atc proteins we identified participate in known mechanisms of cold adaptation, such as membrane composition modification, protein secretion, protein folding or assembly, or if they are involved in any other unanticipated pathways. Therefore, this work paves the way for new insight about stress adaptation mediated by a co-chaperone of the DnaK system and dedicated proteins.

## Methods

**Growth conditions and strains**. The wild-type *S. oneidensis* strain used in this study is MR1-R[46]. Strains were grown aerobically in rich LB medium at the temperatures indicated in the figures. When necessary, LB-rich medium was supplemented with chloramphenicol (25 μg/mL), kanamycin (25 μg/mL), ampicillin (50 μg/mL), or streptomycin (100 μg/mL). The *ΔatcJ*, *ΔatcA*, *ΔatcB*, and *ΔatcC* strains were constructed by homologous recombination as described before[47]. *atcJ* (SO_1850) and *atcC* (SO_1846) coding sequences were deleted entirely. However, since the coding sequences of *atcA* (SO_1849) and *atcB* (SO_1848) overlap, and since the Shine-Dalgarno sequence of *atcC* is located at the 3′ extremity of *atcB*, *atcA* was deleted from base pair 1 to 520 (over 540 bp) and *atcB* from base pair 2 to 747 (over 759 bp). The *E. coli* Δ3 strain (W3110 background) is deleted of the genes coding for the three JDPs *dnaJ*, *cbpA*, and *djlA*[21]. The *E. coli* Bth101 was used for bacterial two-hybrid experiments[27,28]. Transformations in *E. coli* and conjugations in *S. oneidensis* were done as described before[47,48].

**Plasmid constructions**. Plasmids used to obtain the *ΔatcJ*, *ΔatcA*, *ΔatcB*, and *ΔatcC* strains were constructed as follows. A 1 kb sequence composed of two overlapping PCR fragments of 500 bp upstream and downstream of the genes to be deleted was PCR-amplified from the chromosome of *S. oneidensis* MR-1 with specific primers (supplementary Table 1) and cloned into the suicide vector pKNG101[47,49] at the ApaI and SpeI restriction enzyme sites for *atcB* and *atcC*, or BamHI and SpeI for *atcA* and *atcJ*.

To construct plasmids pET-DnaK$_{So}$, pET-DnaJ$_{So}$, pET-GrpE$_{So}$, pET-AtcJ, and pET-AtcC allowing the overproduction of DnaK-6His, DnaJ-6His, 6His-GrpE, AtcJ-6His, and AtcC-6His, respectively, genes were PCR-amplified from the chromosome of *S. oneidensis* MR-1 using forward oligonucleotides with the NdeI restriction site, and reverse nucleotides with the SacI restriction site. To construct plasmid pET-AtcB allowing the overproduction of AtcB-6His, *atcB* was PCR-amplified using forward oligonucleotides with the NdeI restriction site, and reverse nucleotides with the EcoRI restriction site. A sequence coding for a 6-His tag (6xCAC) was added on the forward primer for pET-GrpE$_{So}$, and on the reverse primers for pET-DnaK$_{So}$, pET-DnaJ$_{So}$, pET-AtcJ, pET-AtcB, and pET-AtcC (Supplementary Table 1). After digestion, the DNA fragments were inserted into the pET24b vector (Novagen) digested with the corresponding restriction enzymes. To construct pET52-AtcB and pET52-HtpG$_{So}$ allowing production of Strep-AtcB and Strep-HtpG, the same strategy as above was used except that the PCR products and the pET52 vector (Novagen) were digested with the KpnI and SalI enzymes for *atcB*, and KpnI and SacI enzymes for *htpG*.

For two-hybrid experiments, plasmids pT18-AtcJ, pT18-AtcB, and pT18-AtcC coding for AtcJ, AtcB, and AtcC fused to the C-terminal extremity of the T18 domain of the adenylate cyclase were constructed as described above using the restriction enzymes EcoRI and XhoI and the pUT18-C-linker vector[27]. The pT18-AtcJ-C plasmid was constructed by cloning a synthetic sequence corresponding to the last 21 amino acids of AtcJ in pUT18-C-linker using the restriction sites EcoRI and XhoI. To construct pT25-AtcA, pT25-AtcB, and pT25-AtcC coding for fusion of AtcA, AtcB, and AtcC to the C-terminal extremity of the T25 domain of the adenylate cyclase, the restriction sites used were PstI and XbaI for *atcA*, and EcoRI and XhoI for *atcB* and *atcC*. The DNA fragments were subsequently inserted into pKT25-linker[27].

To construct plasmid pBad33-AtcJ, *atcJ* was amplified by PCR with a forward oligonucleotide containing the XmaI restriction site and a Shine-Dalgarno sequence, and a reverse oligonucleotide containing the XbaI restriction site. After digestion, the PCR product was inserted into the pBad33 vector[50]. The same strategy was used to clone DnaJ from *E. coli* into pBad33 by doing a PCR from *E. coli* MG1655 chromosome using primers containing XmaI and XbaI restriction sites (supplementary Table 1). Plasmids pBad33-AtcA, pBad33-AtcB, and pBad33-AtcC were constructed by digesting the plasmids pET-AtcB and pET-AtcC with XbaI and SalI enzymes, and the digestion products were cloned into the pBad33 vector previously digested with the same enzymes. Therefore, AtcA, AtcB, and AtcC produced from these plasmids contain a 6-His tag at their C-terminal extremity.

To construct pBad33-AtcJ$_{chim}$ coding for the chimera in which the J-domain of DnaJ is replaced by that of AtcJ, a 2-step PCR reaction was performed. First, the sequences coding for the J-domain of AtcJ (amino acids 1 to 73) was amplified by PCR using a reverse primer that overlaps with the forward primer allowing the amplification of the sequence coding for DnaJ$_{Ec}$ without its J-domain (from amino acid 71–376) (Supplementary Table 1). Second, the two overlapping PCR products were mixed and used as matrix for a PCR amplification with external primers containing the XmaI and XbaI restriction enzyme sites, before insertion into the pBad33 vector[50].

For co-purification experiments, pBad24-CBP-AtcB was constructed by digesting pT25-AtcB with XbaI and XhoI enzymes, and the digestion product was cloned into the pBad24-CBP vector[51] previously digested with the same enzymes.

To add the H31Q mutation in the J-domain of AtcJ, plasmids pET-AtcJ, pBad33-AtcJ, and pBad33-AtcJ$_{chim}$ were mutated using the QuickChange kit (Agilent) according to the manufacturer instructions, resulting in plasmids pET-AtcJ$_{H31Q}$, pBad33-AtcJ$_{H31Q}$, and pBad33-AtcJ$_{chimH31Q}$. The same strategy was used to obtain the pET-AtcJ$_{ΔC}$, pT18-AtcJ$_{ΔC}$, and pBad33-AtcJ$_{ΔC}$ plasmids by adding the stop codon TAG instead of AAC coding for N74 by QuickChange reaction. All constructions were verified by sequencing.

**E. coli Δ3 complementation assays**. For complementation of the *E. coli* phenotypes by the chimera, plasmids pBad33-AtcJ$_{chim}$, pBad33-AtcJ$_{chimH31Q}$, or pBad33-DnaJ$_{Ec}$ were introduced in the *E. coli* Δ3 strain (W3110 background) deleted of *dnaJ*, *cbpA*, and *djlA*[21]. Strains were grown at 28 °C until OD$_{600}$ = 3, diluted to OD$_{600}$ = 1, and 10-time serial dilutions were spotted on LB-agar plates supplemented with 2% (w/v) L-arabinose. Plates were then incubated at 37 °C or 43 °C for 24 h. In addition, 2 μL of cells diluted to OD$_{600}$ = 1 were spotted on LB soft agar 0.3% (w/v) supplemented or not with L-arabinose and incubated at 28 °C for 24 h.

**Bacterial growth**. To follow *S. oneidensis* growth in liquid cultures, cells were grown at 7 °C or 28 °C as indicated after overnight precultures at 28 °C, and OD$_{600}$ was measured with time. To follow *S. oneidensis* growth on plates, cells in late exponential growth phase grown at 28 °C were diluted to OD$_{600}$ = 1. Then, 2 μL of 10-time serial dilutions were spotted on LB-agar plates supplemented or not with 0.2% (w/v) L-arabinose. Plates were then incubated at 7 °C for 10 days, 15 °C for 5 days, and 28 °C or 35 °C for 24 h.

**Control of the amount of the AtcJ or AtcJ$_{chim}$ mutants**. To check that the same amount of wild-type and mutant proteins were produced, the *E. coli* Δ3 strains containing the plasmids coding for AtcJ$_{chim}$ wild-type or mutant, and the *S. oneidensis* strains containing the plasmids coding for AtcJ wild-type or mutants, were grown at 28 °C with 0.2% L-arabinose. The total protein extracts from the same amount of cells were heat-denatured, loaded on SDS-PAGE, transferred by western blot and revealed with anti-DnaJ antibody for AtcJ$_{chim}$ wild-type and mutant, anti-AtcJ antibody for AtcJ wild-type and mutants, or anti-HtpG to detect the HtpG protein as a loading control.

**Bacterial Two-hybrid assay**. Bacterial two-hybrid assays were performed as described earlier[27,28] with some modifications. Bth101 strain lacking the adenylate cyclase gene[27,28] was co-transformed with a first plasmid coding for the T18 domain fused to AtcJ wild-type, AtcJ$_{ΔC}$, AtcJ-C, AtcB, or AtcC, and a second plasmid coding for the T25 domain fused to AtcA, AtcB, or AtcC. For negative controls, pUT18-C-linker or pKT25-linker vectors were used[27]. After 2 days at 28 °C, cells were grown overnight in liquid LB medium supplemented with 25 μg/mL kanamycin, 50 μg/mL ampicillin, and 1 mM IPTG. Cells were lysed by adding PopCulture Reagent solution (Agilent) and 1 mg/mL lysozyme during 15 min, before addition of buffer Z (100 mM phosphate buffer pH = 7, 10 mM KCl, 1 mM MgSO$_4$, 50 mM β-mercaptoethanol). Then, 2.2 mM ortho-nitrophényl-β-galactoside (ONPG) was added, and β-galactosidase activity was measured using a modified Miller assay adapted for use in a Tecan Spark microplate reader as described earlier[52].

**Protein purification**. BL21(DE3) strains (Novagen) containing pET-DnaK$_{So}$, pET-GrpE$_{So}$, pET-AtcJ wild-type and mutants, pET-DnaJ$_{So}$, pET52-AtcB, or pET52-HtpG$_{So}$ were cultivated in LB medium at 37 °C until OD$_{600}$ = 0.8. Protein production was induced by adding 1 mM iso-propyl-β-D-thiogalactopyranoside (IPTG), and after 2 h of culture at 28 °C, cells were collected by centrifugation. To produce AtcC, BL21(DE3) strain containing pET-AtcC was grown at 37 °C until OD$_{600}$ = 0.8, 0.1 mM IPTG was added, the cells were incubated at 16 °C for 20 h and collected.

Cells containing His-tag proteins were resuspended in buffer A (40 mM phosphate buffer pH = 7.4, 500 mM NaCl, 10% glycerol), and lysed by French Press. Lysates were centrifuged 10 min at 10,000 rpm, and the supernatants were centrifuged 45 min at 45,000 rpm. The resulting supernatants were then loaded on a His-Trap HP/FF column (GE Healthcare) using an AKTA Pure system (GE Healthcare), the column was washed successively with buffer A containing 5, 20, and 50 mM imidazole. Proteins were eluted with 100, 200, and 500 mM imidazole in buffer A.

Cells containing Strep-AtcB and Strep-HtpG proteins were resuspended in buffer C (100 mM Tris-HCl pH = 8, 150 mM NaCl, 1 mM EDTA) and lysed by French Press. Lysates were centrifuged 10 min at 10,000 rpm, and the supernatants were centrifuged 45 min at 45,000 rpm. The resulting supernatants were then loaded on a Strep-Trap HP column (GE Healthcare) using an AKTA Pure system (GE Healthcare). The column was washed with buffer C, and proteins were eluted in the same buffer containing 2.5 mM desthiobiotin (Sigma-Aldrich).

For 6His- and Strep-tag protein purification, fractions containing the proteins of interest were pooled and concentrated using Amicon Centrifugal Filters (Millipore). DnaK, AtcJ wild-type and mutants, AtcC, and AtcB were purified further using a gel filtration column HiLoad 16/600 Superdex 200 pg (GE Healthcare) in buffer B (50 mM Tris-HCl, 100 mM KCl, 10% glycerol, and 1 mM DTT). GrpE, DnaJ, and HtpG were loaded on a PD-10 column (GE Healthcare) in buffer B. Protein concentrations were measured by Bradford assays. Proteins were stored at −80 °C. The pep21 synthetic peptide corresponding to the last 21 amino acids of AtcC (NNVIVTDPNSAMRELWDQFYP) was obtained from Smart Bioscience.

**ATPase activity assay.** DnaK ATPase activity assays were performed at 37 °C with DnaK (10 μM), AtcJ wild-type or mutant (50 μM), DnaJ (2 μM), and GrpE (2 μM) when indicated, in 25 mM Hepes pH = 7.5, 150 mM KCl, 5 mM MgCl$_2$, 2 mM ATP, using a pyruvate kinase/lactate dehydrogenase enzyme-coupled assay as described before[53].

**Co-purification assays.** Co-purification experiments on cellular extracts were performed as described earlier[16]. Briefly, *E. coli* MG1655 strains containing pBad24-CBP or pBad24-CBP-AtcB with pBad33-AtcC-6His or pBad33 as indicated were grown at 37 °C until OD$_{600}$ = 0.8. Gene expression was induced by adding 0.05% L-arabinose, and cells were collected after 1 h. Cells were then resuspended in buffer D (10 mM Tris-Hcl pH = 8, 150 mM NaCl, 1 mM Mg acetate, 1 mM imidazole, 2 mM CaCl$_2$, 0.1% Triton X-100, 0.1% Tween-20, 20 mM β-mercaptoethanol), and lysed by French Press. After centrifugation, supernatants were incubated for 1 h at 4 °C with calmodulin beads (Agilent). Beads were then washed five times with buffer D and resuspended with loading buffer. After denaturation, proteins were loaded on SDS-PAGE, transferred by western blot, and visualized using anti-CBP (Merck) or anti-His (Thermo) antibodies.

For co-purification experiments using purified proteins, Strep-AtcB (16 μg), Strep-HtpG (16 μg), or AtcC 6-His (100 μg) were mixed as indicated in the figure in buffer E (100 mM Tris-HCl pH = 8, 150 mM NaCl, 1 mM EDTA, 0.05% Triton X-100) before the addition of 50 μL Strep-Tactin beads (IBA Lifesciences). After a 1-h incubation at 4 °C, beads were washed four times with 1 mL of buffer E, and eluted with 20 μL of buffer E supplemented with 5 mM desthiobiotin (Sigma-Aldrich). Proteins were heat denatured and separated by SDS-PAGE subsequently stained with Coomassie Blue.

**Isothermal titration calorimetry.** The AtcJ, AtcJ$_{\Delta C}$, and AtcC proteins were dialyzed overnight at 4 °C in buffer B to avoid discrepancies from buffer mismatch. The pep21 peptide was solubilized in buffer B. ITC experiments were performed at 25 °C using the MicroCal PEAQ-ITC (Malvern). In total, 285 μM of ligands were titrated using 19 injections of 2 μl against 36 μM of AtcC in the ITC cell with a constant stirring speed of 750 rpm, with the exception of pep21, for which only 8 injections were needed to extrapolate the data. Duplicates were performed for each measurement. To account for the dilution heat of the titrated ligands, AtcJ-WT, AtcJ$_{\Delta C}$, and pep21 were titrated, respectively, into buffer and subtracted from the measured data with AtcC. The data were fitted using a "One Set of Sites" model in the PEAQ-ITC Analysis Software.

**Thermal shift assays.** AtcC (10 μM), AtcJ (40 μM), or pep21 (40 μM) were mixed together as indicated in the figure in buffer B (final volume: 20 μL). SYPRO Orange (Sigma-Aldrich) was added to a final concentration of 10X according to the manufacturer, the temperature was increased of 0.5 °C every 30 s from 10 to 90 °C, and fluorescence was measured with time with a BioRad CFX96 Touch Real-Time PCR instrument. Determination of protein melting point was done using BioRad CFX Manager 3.1 software[54].

**RNA preparation, PCR from cDNA, and qRT-PCR.** RNA from *S. oneidensis* wild-type grown at 37 °C, 28 °C, or 7 °C to exponential growth phase (OD$_{600}$ = 2) was extracted and reverse transcribed into cDNA as described earlier for other bacteria[55]. PCR was performed using specific primers (Supplementary Table 1) and cDNA as a matrix. For the negative control, RNA was used as a matrix to make sure that the RNA preparation was not contaminated by the chromosomal DNA.

Amplification was performed using GoTaq® DNA Polymerase (Promega), and the following cycling parameters: 2′ at 95 °C, followed by 29 cycles of 30″ at 95 °C, 30″ at 55 °C, 2′30″ at 72 °C, and a final elongation of 5′ at 72 °C. Samples were loaded on 1% agarose gel prepared in TBE (Tris-Borate-EDTA) buffer and visualized with ethidium bromide staining.

Quantitative real-time PCR (qPCR) analyses were performed on a CFX96 Real-Time System (BioRad) as described before[55]. The results were analyzed using BioRad CFX Manager software, version 3.1 (BioRad, France). The 16s RNA gene was used as a reference for normalization. For each point, technical and biological duplicates were performed.

**Statistics and reproducibility.** Independent experiments were repeated three times to generate replicates used to calculate mean and standard deviation. Where indicated, *t* tests were performed with GraphPad Prism Software version 6.05 to compare two sets of unpaired data using a parametric test with a Welch's correction.

**Reporting summary.** Further information on research design is available in the Nature Research Reporting Summary linked to this article.

## Data availability

All data generated or analyzed during this study are included in this published article (and its supplementary information files). Full-length gels and blots are showed in the Supplementary Fig. 5. All source data underlying the graphs and charts presented in the main figures is made available as Supplementary Data 1.

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

## Acknowledgements

We thank Pierre Genevaux, Marianne Ilbert, Sue Wickner, Aurélia Battesti, Chantal Iobbi-Nivol, Cécile Jourlin-Castelli, Michel Fons and Olivier Lemaire for help and fruitful discussions, Yann Denis for assistance with cDNA preparation and qPCR. Pierre Genevaux, Marie-Pierre Castanié-Cornet, and Latifa El-Antak are thanked for kindly providing strains, antibodies, and peptides. This work was supported by the Centre National de la Recherche Scientifique and Aix Marseille Université (AMU). O.G. was supported by a grant from the Agence Nationale de la Recherche (ANR-16-CE11–0002–01).

## Author contributions

N.J.M., F.A.H., D.B., V.M., and O.G. designed the research, interpreted the data, and wrote the paper. N.J.M., F.A.H., D.B., and O.G. performed experiments.

## Additional information

**Competing interests:** The authors declare no competing interests.

