## [Peer Review File · Communications Biology]

Reviewers' comments:

Reviewer #1 (Remarks to the Author):

Genest and colleagues identified a J-domain protein in the bacterium *Shewanella oneidensis*, AtcJ, that is part of an uncharacterized operon. The authors show that AtcJ behaves like a bona fide J-protein co-chaperone interacting with the molecular chaperone DnaK. Functionally, the AtcJABC operon is shown to be involved in cold resistance and a network of interaction exists between these proteins which is required for their function. The experiments are straight forward and technically sound. The results provide new insight on the DnaK-co-chaperone system that goes significantly beyond the classical paradigm established for *E. coli*.

Specific questions

1. The authors report that AtcJ stimulates the ATPase activity of DnaK by 3 fold. How does this compare to DnaJ? Accordingly, Fig. 1F should also show the ATPase activity of DnaK in the presence of DnaJ. Similarly in Fig. S1C, a direct comparison between DnaK+DnaJ+GrpE and DnaK+AtcJ+GrpE would be useful.
2. In the ATPase assays, the concentration of AtcJ used is quite high i.e. 50 μ M which is 5 times higher than DnaK concentration (10 μ M). What is the KD for DnaK and AtcJ binding?
3. How does the loss of DnaJ and/or DnaK influence cold shock sensitivity?
4. In Fig. S2B, it is shown that DnaK expression is reduced at low temperatures and that AtcJ expression is not increased during cold shock conditions. Are the levels of AtcA, AtcB and AtcC affected by cold shock?
5. To gain more insight into how AtcJABC are related, it could help to check if one component can rescue the loss of other components.
6. In the discussion, the authors hypothesize, that AtcJ may recruit AtcC to DnaK, which then promotes folding. Does loss of DnaK affect AtcC levels?

Reviewer #2 (Remarks to the Author):

Shewanella Oneidensis, an environmental bacteria that is known to convert heavy metals into their elemental form, is shown to have adaptation for cold control. The study was designed to prove that J-domain co-chaperones regulate that. I have two minor suggestions to improve this manuscript.

(i) Figure 2D, should have specific constitutive expressing loading control instead of considering contaminating band as loading control.

(ii) A model figure should be proposed to explain the J-domain co-chaperone dependent regulation of cold adaptation of *Shewanella Oneidensis*.

Reviewer #3 (Remarks to the Author):

The manuscript identifies a previously unknown J-domain protein in *S. oneidensis*, and suggests that it acts in complex with two other proteins to support growth at low temperatures. AtcJ has a J-domain and a short, unique C-terminus. The J-domain replaces that of *E. coli* DnaJ in complementation assays, and AtcJ activates *E. coli* DnaK ATPase with or without GrpE. Deletion of AtcJ causes cold sensitivity in growth, and complementation requires both the active J-domain and the C-terminus.

AtcJ is encoded by a conserved operon with the *atcA*, *B* and *C* genes. Δ *atcB* and *C* had cold sensitivity defects, but not Δ *atcA*. AtcJ and *C* interacted in two-hybrid experiments, requiring the AtcJ C-terminus. ITC measured affinities of AtcJ binding to AtcC to be 7 nM, and of an AtcJ C-terminal peptide to AtcC to be 80 nM. AtcJ or the peptide increased the thermal stability of AtcC. AtcC and *B* also interacted by two-hybrid and pulldown assays. It is hypothesized that AtcC is a misfolded substrate of DnaK, and AtcJ transfers it to DnaK for refolding or remodeling. Alternatively, AtcJ is targeted to AtcJ specific substrates, or localized within the cell by the AtcB-C complex.

The experiments presented are carried out properly, with appropriate controls, and convincing results. The actual mechanisms of how the complex allows cold temperature growth remain unknown, though. The concept that a J-domain protein acts in a complex with other proteins, for specific functions, is not new, and this example is not enough of an advance. There are already several examples in other species, including cases cited in the Discussion (Swa2/auxilin, Zuo1/MPP11; also Sec63 associated with the translocon, *J Cell Sci.* 2014 127:4270-8 and *PLoS One.* 2012; 7:e49243; and RME8/DNAJC13, *J Cell Sci.* 2014 127:2053-70). The interaction between AtcB and AtcC is not shown to be important for biological function, so the essential network is still only the AtcJ-AtcC interaction. *S. oneidensis* is not an established model organism and the findings have limited impact.

Specific points:

1. The binding of AtcB to AtcC was from co-expression in *E. coli*, and it is not certain if the interaction is direct. The idea of a specific network would be strengthened by characterizing the interaction, and showing that it is important for complementation.
2. There is no evidence that AtcJ is misfolded, as proposed in the Discussion. The opposite seems true, because a thermal melting transition is detected in the first place, and the T_m 's of 31°C and 44°C without and with AtcC are above normal growth temperatures. 7 nM binding affinity is quite strong, and also not likely for a misfolded or unstructured protein. Indeed, most transient co-chaperone interactions are weaker, with affinities from 100 nM to μ M, while nM affinity is consistent with stable complexes.
3. In the plasmid complementation experiments, the expression levels from the plasmids should be shown to be comparable to those of endogenous proteins.
4. If the AtcJ-AtcC-AtcB network is required for function, then high overexpression of AtcC or AtcB should also not suppress the Δ *atcJ* defect.
5. Stoichiometry of binding is measured in the ITC experiments in Fig. 4 and should be reported. A peptide with a scrambled sequence as a control will show that binding is sequence specific.

Response to reviewers and editor

Authors: We would like to thank the reviewers and the editor for having carefully reviewed our manuscript. We have responded to the points raised by the reviewers. Below is a point by point response to the reviewer comments.

Reviewer #1:

Genest and colleagues identified a J-domain protein in the bacterium *Shewanella oneidensis*, AtcJ, that is part of an uncharacterized operon. The authors show that AtcJ behaves like a bona fide J-protein co-chaperone interacting with the molecular chaperone DnaK. Functionally, the AtcJABC operon is shown to be involved in cold resistance and a network of interaction exists between these proteins which is required for their function. The experiments are straight forward and technically sound. The results provide new insight on the DnaK-co-chaperone system that goes significantly beyond the classical paradigm established for *E. coli*.

Authors: We would like to thank the reviewer for his positive feedback.

1. The authors report that AtcJ stimulates the ATPase activity of DnaK by 3 fold. How does this compare to DnaJ? Accordingly, Fig. 1F should also show the ATPase activity of DnaK in the presence of DnaJ. Similarly in Fig. S1C, a direct comparison between DnaK+DnaJ+GrpE and DnaK+AtcJ+GrpE would be useful.

Authors: We measured the stimulation of the ATPase activity of DnaK by DnaJ. We found that DnaJ stimulated about 6 times the ATPase activity of DnaK at a 1:0.2 stoichiometry (DnaK:DnaJ). This result is similar to what has been measured before with the DnaK and DnaJ proteins from *E. coli* (for example Diamant and Goloubinoff, *Biochemistry* 1998, 9688-9694). Since we found that AtcJ stimulated the ATPase activity of DnaK about three fold at a 1:5 stoichiometry (DnaK:AtcJ) (Fig. 1F), it indicates that DnaJ is more efficient than AtcJ for stimulation. In the presence of GrpE, higher stimulation was also measured in the presence of DnaJ compare to AtcJ. These results have been added in the figure 1F and supplementary figure 1C.

The lower stimulation of the DnaK ATPase activity by AtcJ compare to DnaJ was expected since it has been shown that in addition to the J-domain, other domains of DnaJ (absent in

AtcJ) including the G/F region participate in the stimulation (Wall, Zylicz and Georgopoulos, JBC 1994, 5446-51; Kityk, Kopp and Mayer, Mol. Cell 2018, 227-237).

2. In the ATPase assays, the concentration of AtcJ used is quite high i.e. 50 μ M which is 5 times higher than DnaK concentration (10 μ M). What is the KD for DnaK and AtcJ binding?

Authors: To determine binding constants, we tried several in vitro approaches including ITC and DLS. We were not able to get clear data showing binding between DnaK and AtcJ despite many attempts in different conditions. Given that AtcJ is a very short JDP protein composed of a J-domain and a 21-amino-acid extension involved in AtcC interaction, this result is not surprising. Indeed, it appears from the literature that binding between J-domains alone and DnaK has been extremely difficult to study because the interaction is highly transient (Mayer et al., J. Mol. Biol. 1999, 1131-44; Greene et al., 1998 PNAS, 6108-13). Recently, to overcome this weak binding and get structural information, Mayer and colleagues had to fuse the J-domain of DnaJ to DnaK (Kityk, Kopp and Mayer, Mol. Cell 2018, 227-237). Interaction with full-length DnaJ and DnaK is easier to measure because binding to DnaK is stabilized by contacts between the additional domains of DnaJ (absent in AtcJ) and the substrate binding region of DnaK (Mayer et al., J. Mol. Biol. 1999, 1131-44; Laufen et al., PNAS 1999, 5452-7).

In vivo, we used a bacterial two-hybrid assay (see below) that indicates that AtcJ and DnaK interact. The interaction with DnaK is most probably specific since there was a significant decrease in binding when AtcJ_{H31Q} (mutation in the conserved HPD tripeptide involved in DnaK binding) was used instead of AtcJ. Although it is impossible to determine affinity constants by this approach, a stronger level of interaction was measured between DnaJ and DnaK compare to AtcJ and DnaK. These results are in agreement with the literature cited above and one possibility is that in the in vivo bacterial two-hybrid assay, a protein interacting as a substrate with DnaK could stabilize the interaction between AtcJ and DnaK.

DnaJ and AtcJ interact with DnaK based on bacterial two-hybrid assays. *E. coli* Bth101 strain co-transformed as indicated with the T18 and T25 plasmids coding for protein fusion between T18 and DnaK, and protein fusion between T25 and DnaJ or AtcJ wild type or mutant in the conserved HPD tripeptide involved in DnaK binding were grown overnight at 28°C. β -galactosidase activity was measured as explained in Methods. Data from three replicates are shown as mean \pm SD.

3. How does the loss of DnaJ and/or DnaK influence cold shock sensitivity?

Authors: According to global studies using transposon insertion followed by sequencing in *S. oneidensis* (Yang et al., PloS Computational Biology 2014, e1003848), the *dnaK* and *dnaJ* genes were indicated as essential. Nevertheless, we experimentally tried to construct the deletion mutants, but were unsuccessful, confirming that these two genes are indeed essential. We therefore cannot test the effects of the loss of DnaJ and/or DnaK on cold shock sensitivity.

4. In Fig. S2B, it is shown that DnaK expression is reduced at low temperatures and that AtcJ expression is not increased during cold shock conditions. Are the levels of AtcA, AtcB and AtcC affected by cold shock?

Authors: qPCR experiments were performed to answer this point. We found that the levels of expression of *atcA*, *atcB*, and *atcC* were similar to the level of expression of *atcJ* at 7°C, 28°C and 37°C. A very slight increase in the expression level of the operon is observed at low temperature. These results are in accordance with the fact that *atcJ*, *atcA*, *atcB* and *atcC* are encoded in the same operon. These results have been added in supplementary figure 3 and in the result section (lines 200 to 206).

5. To gain more insight into how AtcJABC are related, it could help to check if one component can rescue the loss of other components.

Authors: We performed the experiment suggested by the reviewer and looked at the growth at low temperature of the *atcJ*, *atcB*, and *atcC* mutant strains that produced the other Atc proteins. We found that the low growth observed in the absence of *atcJ*, *atcB*, or *atcC* was not rescued by the production of the other components. It suggests that each Atc protein plays a distinct role to support growth in cold condition and cannot be replaced by the other Atc proteins. These results have been added in supplementary figure 2. The same point was also raised by Reviewer 3 (point 4)

6. In the discussion, the authors hypothesize, that AtcJ may recruit AtcC to DnaK, which then promotes folding. Does loss of DnaK affect AtcC levels?

We agree with the reviewer that it would be an appealing model to show that the level of AtcC would be reduced in the absence of DnaK. However, we unfortunately cannot respond to this point since DnaK is essential in *S. oneidensis* (see also point 3).

Reviewer #2

Shewanella Oneidensis, an environmental bacteria that is known to convert heavy metals into their elemental form, is shown to have adaptation for cold control. The study was designed to prove that J-domain co-chaperones regulate that. I have two minor suggestions to improve this manuscript.

Authors: We would like to thank the reviewer for his positive feedback.

(i) Figure 2D, should have specific constitutive expressing loading control instead of considering contaminating band as loading control.

Authors: We understand the reviewer concern about the loading control we used in the initial figure. As a loading control, we now looked at the HtpG protein using an antibody directed against this protein. The figure 2D has been modified accordingly.

(ii) A model figure should be proposed to explain the J-domain co-chaperone dependent regulation of cold adaptation of Shewanella Oneidensis.

Authors: We have added a model in figure 6. This model is explained in the discussion section of the manuscript (lines 326 to 334)

Reviewer #3:

The manuscript identifies a previously unknown J-domain protein in *S. oneidensis*, and suggests that it acts in complex with two other proteins to support growth at low temperatures. AtcJ has a J-domain and a short, unique C-terminus. The J-domain replaces that of *E. coli* DnaJ in complementation assays, and AtcJ activates *E. coli* DnaK ATPase with or without GrpE. Deletion of AtcJ causes cold sensitivity in growth, and complementation requires both the active J-domain and the C-terminus. AtcJ is encoded by a conserved operon with the *atcA*, *B* and *C* genes. Δ *atcB* and *C* had cold sensitivity defects, but not Δ *atcA*. AtcJ and *C* interacted in two-hybrid experiments, requiring the AtcJ C-terminus. ITC measured affinities of AtcJ binding to AtcC to be 7 nM, and of an AtcJ C-terminal peptide to AtcC to be 80 nM. AtcJ or the peptide increased the thermal stability of AtcC. AtcC and B

also interacted by two-hybrid and pulldown assays. It is hypothesized that AtcC is a misfolded substrate of DnaK, and AtcJ transfers it to DnaK for refolding or remodeling. Alternatively, AtcJ is targeted to AtcJ specific substrates, or localized within the cell by the AtcB-C complex.

The experiments presented are carried out properly, with appropriate controls, and convincing results. The actual mechanisms of how the complex allows cold temperature growth remain unknown, though. The concept that a J-domain protein acts in a complex with other proteins, for specific functions, is not new, and this example is not enough of an advance. There are already several examples in other species, including cases cited in the Discussion (Swa2/auxilin, Zuo1/MPP11; also Sec63 associated with the translocon, J Cell Sci. 2014 127:4270-8 and PLoS One. 2012; 7:e49243; and RME8/DNAJC13, J Cell Sci. 2014 127:2053-70). The interaction between AtcB and AtcC is not shown to be important for biological function, so the essential network is still only the AtcJ-AtcC interaction. *S. oneidensis* is not an established model organism and the findings have limited impact.

Authors: We would like to thank the reviewer for his comments.

Specific points:

1. The binding of AtcB to AtcC was from co-expression in *E. coli*, and it is not certain if the interaction is direct. The idea of a specific network would be strengthened by characterizing the interaction, and showing that it is important for complementation.

Authors: We have added a new experiment in the paper that indicates that the binding we observed between AtcB and AtcC in the bacterial two-hybrid assay and the in vivo pull-down is indeed direct. For that, AtcC with a 6-His tag and AtcB with a Strep tag were purified to homogeneity (at least 95% pure). We then performed a co-purification in vitro on purified proteins. We found that AtcC co-purified with AtcB. As an additional control, we checked that the binding between AtcB and AtcC is specific since AtcC did not co-purify with another unrelated protein (HtpG with a Strep tag). These results have been added in Figure 5C.

We agree with the reviewer that identifying point mutants of AtcB or AtcC that prevent binding between the two proteins and showing that these mutants do not support growth at low temperature would strengthen the idea of a specific network. However, these experiments are far beyond the scope of our study; we hope we will present these data in a near future.

2. There is no evidence that AtcJ is misfolded, as proposed in the Discussion. The opposite seems true, because a thermal melting transition is detected in the first place, and the T_m 's of 31°C and 44°C without and with AtcC are above normal growth temperatures. 7 nM binding affinity is quite strong, and also not likely for a misfolded or unstructured protein. Indeed, most transient co-chaperone interactions are weaker, with affinities from 100 nM to μ M, while nM affinity is consistent with stable complexes.

Authors: We thank the reviewer for this valuable comment. We indeed do not have evidence to show that AtcC is misfolded at low temperature. Since reviewer 2 asked for a model (now in figure 6), we have modified this part of the discussion (lines 326 to 334) to reflect the hypotheses suggested by our data.

3. In the plasmid complementation experiments, the expression levels from the plasmids should be shown to be comparable to those of endogenous proteins.

Authors: In *Shewanella oneidensis*, the variety of plasmids we can introduce is limited since they have to possess a pACYC replication origin. We used the pBad33, which is a low copy vector (Guzman et al., J. Bacteriol. 1995, 4121-4130). Its expression can be tightly controlled by adding the arabinose inducer, in the range of 0% (expression leak) to 2% (high expression). We titrated down the level of arabinose (0.2%, 0.02% and 0%) in our complementation experiments at 7°C. As you can see on supplementary figure 2, we found that 0.02% arabinose was sufficient to allow complementation of the growth phenotype of the $\Delta atcB$ /pAtcB, $\Delta atcC$ /pAtcC, and $\Delta atcJ$ /pAtcJ strains. Interestingly, we found that even without arabinose (leaky expression) there was enough AtcB and AtcC to rescue the growth of the $\Delta atcB$ and $\Delta atcC$ strains, respectively.

We then wanted to estimate the level of expression from the plasmid compare to the chromosomal level. To do that, the WT/p, $\Delta atcB$ /pAtcB, $\Delta atcC$ /pAtcC, and $\Delta atcJ$ /pAtcJ strains were grown at 7°C in the presence of 0.02% arabinose until late exponential phase, total RNA were extracted and qPCR were performed using specific primers. We found that there was about 20 times more expression from the plasmid than from the chromosome (see below). Therefore, it indicates that, although there is more expression from the plasmids than from the chromosome (as expected with a plasmid), there is for sure no need of a massive overproduction of the proteins (using less than 0.02% for AtcJ, and even no arabinose for AtcB and AtcC) to complement the growth phenotype.

Level of expression from the plasmids compare to chromosomal level. *S. oneidensis* strains as indicated were grown at 7°C in the presence of 0.02% arabinose. The level of expression of *atcB*, *atcC*, and *atcJ* were quantified by RT-qPCR and are expressed relatively to the level of chromosomal expression. Data from two replicates are shown as mean \pm SEM.

4. If the AtcJ-AtcC-AtcB network is required for function, then high overexpression of AtcC or AtcB should also not suppress the $\Delta atcJ$ defect.

Authors: We performed the experiment suggested by the reviewer by using 0.2% arabinose and looked at the growth at low temperature of the *atcJ*, *atcB*, and *atcC* mutant strains that produced the other Atc proteins. We found that the low growth observed in the absence of *atcJ*, *atcB*, or *atcC* was not rescued by the production of the other components. It suggests that each Atc protein plays a distinct role to support growth in cold condition and cannot be replaced by the other Atc proteins. These results have been added in supplementary figure 2. The same point was also raised by Reviewer 1 (point 5).

5. Stoichiometry of binding is measured in the ITC experiments in Fig. 4 and should be reported. A peptide with a scrambled sequence as a control will show that binding is sequence specific.

Authors: We apologized for not having mentioned the 1:1 stoichiometry of binding in Figure 4 for AtcJ-AcC binding. It has now been added in the text (line 257).

To show that binding is sequence specific, we tested the interaction of AtcC with a 24-amino-acid control peptide (SRSSLASAWGRFLLQRGSWTGPRC). We found that this peptide did not interact with AtcC. This control has been added in supplementary figure 4.

REVIEWERS' COMMENTS:

Reviewer #1 (Remarks to the Author):

the authors have answered my queries in a highly satisfactory manner. The new data further strengthen the story.

Reviewer #2 (Remarks to the Author):

The Authors in this manuscript have attempted to show the mechanistics of role of J. domain of *Shewanella oneidensis*' a bacteria that converts heavy metals into the elemental form.

They have further dissected out the role of important proteins notably J. domain chaperone in this process. The manuscript was reviewed by 3 independent reviewers. While one reviewer has asked for minor modifications, other two reviewers have asked for major revision. The revised version tries to address all the issues raised by the Reviewers. One of the Reviewer asked for comparison of ATPase activity stimulation between DnaK and DnaJ. The Authors have been able to carry out this experiment and have incorporated the same in the Fig 1f and also in supplementary Fig 1c. The issue of binding constant for DnaK/DnaJ with AtcA, AtcB and AtcV was raised to which the Authors have carried out new experiments based on bacterial 2 hybrid assays. The question of loss of DnaJ/DnaK influence on cold shock sensitivity could not be answered since it has been earlier suggested that these are essentials. The Authors tried to construct deletion mutants which was unsuccessful, confirming earlier studies that these 2 proteins are essential proteins.

The Authors have added supplementary fig 2 to include results on the levels of AtcA, AtcB and AtcC as a function of cold shock. They have also added supplementary fig 2 to show this.

To answer the issue of whether binding of AtcB to AtcC is a direct interaction the authors carried out new experiments using bacterial 2 hybrid system and in-vivo assays and these have been included in fig 5C.

Since they are now unable to provide evidence for misfolding of AtcJ as mentioned in the original discussion, they have now modified this in the discussion and also summarized the same in fig 6.

Lastly, the stoichiometry of binding was measured in ITC experiment and the same has been mentioned in the revised manuscript.

The minor modifications suggested by one of the Reviewers have all been carried out by modifying fig 2 and also by adding fig 6.

In view of the extensive revision carried out by the Authors convincingly answering all the issues raised by the reviewers.

I recommend this manuscript for publication.

Reviewer #3 (Remarks to the Author):

The manuscript is improved by the new data.

1. The new Fig. 5c shows interaction by pull-down between purified AtcB and AtcC. The AtcC band is faint but visible, and not present in the controls. This strengthens the idea of a network of direct interactions. It still remains to be determined how important the interaction is for function.
2. Supplemental Fig. 2 shows that Δ atcJ is complemented by atcJ on a plasmid at the low arabinose 0.02%, and Δ atcB and Δ atcC by their respective genes on plasmids even without induction. Based on mRNA, low induction is around 20-fold above endogenous levels. The results show that each Atc gene can only be complemented by itself, not by the others at high expression levels, which is an important point. The uninduced (leaky) levels of atcB and C mRNA were not measured, but must be less than 20-fold. There is still some concern about the high levels of atcJ in the complementation, and it seems odd that the anti-AtcJ antibody was not used for a Western blot. However, the other genes cannot substitute for it.
3. The scrambled peptide control for the ITC is appropriate.
4. The Discussion recognizes that AtcB and C may not be substrates of DnaK.